# Optimized CSIDH Implementation Using a 2-Torsion Point

**Donghoe Heo [1] , Suhri Kim [1], Kisoon Yoon [2], Young-Ho Park [3] and Seokhie Hong [1],\***

[1]  Graduate School of Information Security, Institute of Cyber Security and Privacy (ICSP), Korea University, Seoul 02841, Korea; dong5641@korea.ac.kr (D.H.); suhrikim@gmail.com (S.K.)
[2]  NSHC Inc., Seoul 08502, Korea; kisoon.yoon@gmail.com
[3]  Department of Information Security, Graduate School of Information Security, Sejong Cyber University, Seoul 05000, Korea; youngho@sjcu.ac.kr
[*]  Correspondence: shhong@korea.ac.kr; Tel.: +82-10-6201-6348

**Abstract:** The implementation of isogeny-based cryptography mainly use Montgomery curves, as they offer fast elliptic curve arithmetic and isogeny computation. However, although Montgomery curves have efficient 3- and 4-isogeny formula, it becomes inefficient when recovering the coefficient of the image curve for large degree isogenies. Because the Commutative Supersingular Isogeny Diffie-Hellman (CSIDH) requires odd-degree isogenies up to at least 587, this inefficiency is the main bottleneck of using a Montgomery curve for CSIDH. In this paper, we present a new optimization method for faster CSIDH protocols entirely on Montgomery curves. To this end, we present a new parameter for CSIDH, in which the three rational two-torsion points exist. By using the proposed parameters, the CSIDH moves around the surface. The curve coefficient of the image curve can be recovered by a two-torsion point. We also proved that the CSIDH while using the proposed parameter guarantees a free and transitive group action. Additionally, we present the implementation result using our method. We demonstrated that our method is 6.4% faster than the original CSIDH. Our works show that quite higher performance of CSIDH is achieved while only using Montgomery curves.

**Keywords:** post-quantum cryptography; isogeny; Montgomery curves; two-torsion points; Commutative Supersingular Isogeny Diffie-Hellman (CSIDH)

---

## 1. Introduction

With the evolution of a quantum computing environment, currently used public key cryptosystems based on factorization and discrete logarithm problems, such as RSA and ECC, will not be able to guarantee their security in the near future. This has led to the need for post-quantum cryptography (PQC) that is secure, even in quantum computing environments. The National Institute of Standards and Technology (NIST) opened the PQC standardization project, which is now in Round 2. Among the PQC categories, isogeny-based cryptography interests many researchers, as it offers smaller key sizes than any other PQC candidates. The isogeny-based cryptography is based on the difficulty of finding a specific isogeny between two elliptic curves defined on the same finite field. Despite having a fairly small key size, isogeny-based cryptography has the disadvantage of being considerably slower than most of the PQC candidates.

The isogeny-based cryptography was first proposed by Couveignes in 2006 [1]. This is a non-interactive key exchange protocol, which uses a set of $\mathbb{F}_q$-isomorphism classes of ordinary elliptic curves that are defined on $\mathbb{F}_q$. The endomorphism ring between these curves is given by the order $\mathcal{O}$ in an imaginary quadratic field. Subsequently, the ideal class group $\mathrm{cl}(\mathcal{O})$ acts freely and transitively on

this endomorphism ring through an isogeny operation. Couveignes designed the Diffie-Hellman style key exchange protocol using the commutativity of $cl(\mathcal{O})$. This method was rediscovered by Rostovtsev and Stolbunov and is called CRS-scheme. On the other hand, the underlying problem of CRS-scheme can be classified as an abelian hidden-shift problem. It is shown by Childs et al. that there is a subexponential quantum attack algorithm with time complexity of $L_q[1/2]$ [2]. When considering that RSA is widely used, even in subexponential complexity in classical computers this was not considered to be a big problem. However, very slow execution time was pointed out as the biggest problem, as it took several minutes for a single key exchange.

The isogeny-based cryptography was noticed again with a rapid speed improvement by De Feo et al. [3]. They proposed a new key exchange protocol, called SIDH, while using a supersingular curve. As Childs-Jao-Soukharev's attack exploits the commutativity of $cl(\mathcal{O})$ of an ordinary curve, their attack cannot be applied to SIDH since it uses supersingular curves, which have non-commutative full endomorphism ring. Until now, the best known attacks against SIDH have exponential time complexity, even in quantum computing environments.

SIKE (Supersingular Isogeny Key Encapsulation), which is based on SIDH, is currently on the NIST PQC standardization Round 2 [4]. On the other hand, in the case of SIDH-based scheme, the key validation problem could not be efficiently solved. To solve this problem, SIKE applied a transformation that was similar to the Fujisaki–Okamoto transformation proposed in [5].

In the CRS-scheme, efficient key validation is possible, so that CCA-secure encryption can only be achieved by the basic algorithm itself, without the need of applying FO-transformation. This allows for a non-interactive key exchange, where several of the previously proposed PQC algorithms do not efficiently provide this property. With this in mind, De Feo et al. proposed a method to efficiently perform CRS-schemes on ordinary curves in [6]. However, there was still a problem that it was difficult to select parameters that satisfy a certain condition because of the characteristics of ordinary curves. Independently, Castryck et al. proposed CSIDH (Commutative Supersingular Isogeny Diffie-Hellman), an algorithm that increases efficiency over conventional techniques by using the supersingular curve defined over a prime field $\mathbb{F}_p$ in the CRS-scheme [7]. By using supersingular curves, CSIDH solved the parameter selection problem of ordinary curves in the algorithm proposed by De Feo et al.

CSIDH uses a subring that consists of $\mathbb{F}_p$-rational endomorphisms instead of using a full endomorphism ring, and it uses the commutativity of $cl(\mathcal{O})$ and has the same protocol as CRS-scheme. The CSIDH-512 provides a key size of 64 bytes, which is smaller than SIKE for the same security level. Even when considering the subexponential time attack, the key size is expected to be relatively smaller than SIKE. Recently, various papers that were related to CSIDH have been submitted to PQCrypto 2019 and Eurocrypt 2019, and the various researches, such as digital signature, efficient implementation techniques, various attack techniques, and side-channel resistant implementations, have been conducted [8–11].

However, one disadvantage of CSIDH is that it has a slower execution speed than the state-of-the-art implementation of SIKE. On the other hand, since the key validation can be performed efficiently, a non-interactive key exchange can be provided, and a smaller key size and a simpler algorithm can be designed. In addition, when considering a more efficient digital signature scheme than SIDH can be derived, it is possible to say that CSIDH has more potential for developing various cryptographic applications. Hence, various studies are being actively conducted to improve the speed of CSIDH [8,9].

The original implementation of CSIDH in [7] uses Montgomery curves, as they were known to provide efficient isogeny computation. However, one drawback of using Montgomery curves is that the computational cost for recovering the coefficient of the image curve is higher than Edwards curves for large degree isogenies. Because tge CSIDH protocol uses large odd-degree isogenies, this can be an obstacle for CSIDH to entirely implement on Montgomery curves.

In this paper, we apply an optimization technique that was proposed by Costello and Hisil in CSIDH in order to obtain image curve coefficients during isogeny computations [12]. The following are the main contributions of this work.

- We present a new initial curve and a new prime of the form $8k + 7$, enabling the use of the two-torsion method by Costello and Hisil [12]. In the parameter presented in the original CSIDH, $\mathbb{F}_p$-rational two-torsion points do not exist, except for $(0,0)$, so that this method cannot be used for recovering the coefficient of the image curve in CSIDH. Compared to Meyer's method [8], computing the coefficient of the image curve is the main bottleneck for implementing faster CSIDH entirely on Montgomery curves. By using our prime, $\mathbb{F}_p$-rational two-torsion points exist, so that the coefficient can be efficiently computed.
- We also prove that our algorithm assures one-to-one correspondence between image curves and elliptic curve isomorphism classes. Given a Montgomery curve $M_A : y^2 = x^3 + Ax^2 + x$ on the surface with curve coefficient $A$ and base field prime $p$, we prove that the ideal-class group $\mathrm{cl}(\mathcal{O})$ acts freely and transitively on the set $S^+_{p,\mathbb{Z}[(1+\sqrt{-p})/2],i}$ in [13]. The details of our proof are denoted in Section 4.
- We present the implementation results of our proposed method. The group action of our implementation is about 7.1% faster than the original CSIDH. The entire key exchange is about 6.4% faster than the original CSIDH. Although the proposed CSIDH implementation is slower than [8], we stress the fact that we provide the fastest performance using only Montgomery curves. Section 5 denote details of our implementation and results.

This paper is organized, as follows. In Section 2, we review on background of elliptic curves and CSIDH key exchange. In Section 3, we introduce the various way of odd-degree isogeny computations. In Section 4, we present a new parameter that makes the use of the two-torsion point and our optimization methods. Section 5 describes the specific implementation process and the result of comparing the costs and speed. We draw our conclusions and future work in Section 6.

## 2. Preliminary

In this section, we describe the background knowledge needed to develop this paper. First, we review some properties of elliptic curves. Subsequently, we introduce the CSIDH protocol.

### 2.1. Elliptic Curves and Isogenies

#### 2.1.1. Montgomery Curves

Let $K$ be a field with the characteristic not equal to 2 or 3. The Montgomery elliptic curves over $K$ are expressed by the following equation:

$$M_{a,b} : by^2 = x^3 + ax^2 + x,$$

where $b(a^2 - 4) \neq 0$. We shall write $M_a$ when $b = 1$ throughout the paper. For efficient implementation of isogeny operation, we use the projective coordinate and projective curve coefficient to avoid inversions. Because Montgomery curve arithmetic can only be constructed with the $x$-coordinate, $XZ$-coordinate system is mainly used for implementing isogeny-based cryptography. Now, we write a point $P = (x, y)$ on $M_{a,b}$ and coefficient $a$ as $P = (X : Z)$ and $a = (A : C)$, respectively, where $x = X/Z$ and $a = A/C$.

#### 2.1.2. Isogeny

Let $O_E$ be the group identity of a group of an elliptic curve $E$. Given two elliptic curves $E$ and $E'$, we define an isogeny $\phi$ between $E$ and $E'$ by $\phi : E \to E'$ satisfying $\phi(O_E) = O_{E'}$, where $\phi$ is a morphism.

Because $\phi$ is group homomorphism between $E$ and $E'$, $\ker(\phi)$ is a subgroup of $E$. Given any finite subgroup $K$ of $E$, we use Velu's formula to compute an isogeny $\phi : E \to E'$. Subsequently, we obtain an isogeny $\phi : E \to E'$ satisfying $\ker(\phi) = K$ and denote $\deg(\phi) = |K|$.

### 2.1.3. Supersingularity

Given a prime $p$, let $E$ be an elliptic curve defined over $\mathbb{F}_p$. Afterwards, $E$ is a supersingular curve if and only if

$$\#E(\mathbb{F}_p) = p + 1$$

Otherwise, $E$ is an ordinary curve. Let $\text{End}(E)$ be a full endomorphism ring of $E$ and $\text{End}_{\mathbb{F}_p}(E)$ be an $\mathbb{F}_p$-rational endomorphism ring defined over $\mathbb{F}_p$. A full endomorphism ring of an ordinary curve is isomorphic to an order in an imaginary quadratic field. On the other hand, A full endomorphism ring $\text{End}(E)$ of a supersingular curve $E$ is isomorphic to an order in a quaternion algebra. Additionally, $\mathbb{F}_p$-rational endomorphism ring $\text{End}_{\mathbb{F}_p}(E)$ of supersingular curve $E$ is isomorphic to an order in an imaginary quadratic field $\mathbb{Q}(\sqrt{-p})$. Now, denote an order $\mathcal{O}$ for $\text{End}_{\mathbb{F}_p}(E)$.

### 2.1.4. Ideal Class Group

Given an order $\mathcal{O}$, the ideal class group $\text{cl}(\mathcal{O})$ is defined by a quotient group

$$\text{cl}(\mathcal{O}) = I(\mathcal{O})/P(\mathcal{O})$$

Note that $I(\mathcal{O})$ is the set of invertible fractional ideals and $P(\mathcal{O})$ is the set of principal fractional ideals.

Let $\pi \in \mathcal{O}$ be the $\mathbb{F}_p$-Frobenius endomorphism of $E$ and $\mathcal{E}\ell\ell_p(\mathcal{O}, \pi)$ be the set of elliptic curves $E$ defined over $\mathbb{F}_p$ satisfying $\mathcal{O} = \text{End}_{\mathbb{F}_p}(E)$. Afterwards, the ideal-class group $\text{cl}(\mathcal{O})$ acts freely and transitively on $\mathcal{E}\ell\ell_p(\mathcal{O}, \pi)$ by

$$\text{cl}(\mathcal{O}) \times \mathcal{E}\ell\ell_p(\mathcal{O}, \pi) \longrightarrow \mathcal{E}\ell\ell_p(\mathcal{O}, \pi)$$
$$([\mathfrak{a}], E) \longrightarrow E/\mathfrak{a}$$

### 2.2. Commutative Supersingular Isogeny Diffie-Hellman (CSIDH)

### 2.2.1. CSIDH Protocol

CSIDH is an isogeny-based Diffie-Hellman protocol proposed by Castryck et al. [7] using supersingular curves defined over $\mathbb{F}_p$ and commutative group action. The prime $p$ of the base field is of the form $p = 4 \prod_{i=1}^{n} \ell_i - 1$, where $\ell_i$'s are odd primes. For an order $\mathcal{O} = \text{End}_{\mathbb{F}_p}(E)$, it is well-known that the class group $\text{cl}(\mathcal{O})$ acts freely and transitively on $\mathcal{E}\ell\ell_p(\mathcal{O})$. This group action is represented by $[\mathfrak{a}]E$, where $E \in \mathcal{E}\ell\ell_p(\mathcal{O})$ and an ideal class $[\mathfrak{a}] \in \text{cl}(\mathcal{O})$. Since $E$ is a supersingular curve with $\#E(\mathbb{F}_p) = p + 1 = 4 \cdot \ell_1 \cdots \ell_n$, for each $i$, there is $\mathbb{F}_p$-rational subgroup of order $\ell_i$. Additionally, let $\pi = \sqrt{-p}$ be the $\mathbb{F}_p$-Frobenius endomorphism of $E$. Subsequently, since $p = -1 \mod \ell_i$, for a prime $\ell_i$, it is well-known that $\ell_i \mathcal{O}$ splits into two prime ideals $\mathfrak{l}_i = (\ell_i, \pi - 1)$ and $\mathfrak{l}_i^{-1} = (\ell_i, \pi + 1)$. Using Velu's formula, we compute $[\mathfrak{l}_i]E$ through the isogeny $\phi_{\mathfrak{l}_i}$ defined over $\mathbb{F}_p$ and compute $[\mathfrak{l}_i^{-1}]E$ through the isogeny $\phi_{\mathfrak{l}_i^{-1}}$ defined over $\mathbb{F}_{p^2}$.

Assume that Alice and Bob execute a key exchange. Alice and Bob randomly select each secret key $[\mathfrak{a}]$ and $[\mathfrak{b}]$ in $\text{cl}(\mathcal{O})$, respectively. Next, Alice sends $E_A = [\mathfrak{a}]E$ to Bob, Bob sends $E_B = [\mathfrak{b}]E$ to Alice. Upon the receipt of $E_B$ from Bob, Alice computes $[\mathfrak{a}]E_B$ and obtains $E_{AB} = [\mathfrak{a}]E_B$. Similarly, Bob obtains $E_{BA} = [\mathfrak{b}]E_A$. The $E_{AB} = E_{BA}$ is the shared secret between Alice and Bob.

2.2.2. CSIDH Group Action

An element of the ideal-class group $\text{cl}(\mathcal{O})$ is of the form $\prod_{i=1}^{n} \mathfrak{l}_i^{e_i}$ ($\mathfrak{l}_i = (\ell_i, \pi - 1)$) for small $e_i \in [-m, m]$. Accordingly, in CSIDH protocol, Alice and Bob randomly select a vector $(e_1, e_2, \cdots, e_n) \in \mathbb{Z}^n$ and consider it as a secret key. Thus, a group action $[\mathfrak{a}]E$ can be computed by applying $\ell_i$-isogeny operation $e_i$ times for $\mathfrak{a} = \prod_{i=1}^{n} \mathfrak{l}_i^{e_i} \in \text{cl}(\mathcal{O})$.

If $e_i > 0$, $\ell_i$-isogeny is applied with the kernel generated by a point in $E(\mathbb{F}_p)$ of order $\ell_i$. If $e_i < 0$, $\ell_i$-isogeny is applied with the kernel generated by a point in $E(\mathbb{F}_{p^2} \backslash \mathbb{F}_p)$ of order $\ell_i$. As $\ell_i$s are all primes, this means that efficient odd-degree isogeny formula at least up to 587 for CSIDH-512 is required for implementation. For Montgomery curves, Costello and Hisil proposed an efficient method for computing odd-degree isogenies [12]. For twisted Edwards curves, Moody and Shumow proposed generalized odd-degree isogeny formula [14]. In [15], they optimized the Moody and Shumow formula by using the $w$-coordinate on Edwards curves.

## 3. Odd-Degree Isogenies

Generally, an isogeny operation is divided into two parts—evaluation of an isogeny and coefficients computation of an image curve. In this section, we shall briefly introduce the formula in [12] for point evaluations. For coefficient computations, we introduce various methods that can be used to implement CSIDH. From this section, **M**, **S**, and **a** refer to a field multiplication, squaring, and addition, respectively.

*3.1. Point Evaluation*

In [12], Costello and Hisil proposed a simple formula for computing arbitrary degree isogenies on Montgomery curves. Their formula can be summarized as follows.

**Theorem 1.** *For a field K, whose characteristic is not 2, let P be a point of order $\ell = 2d + 1$ on the Montgomery curve $M_{a,b}/K : by^2 = x^3 + ax^2 + x$. Writing $\sigma = \sum_{i=1}^{d} x_{[i]P}$, $\tilde{\sigma} = \sum_{i=1}^{d} 1/x_{[i]P}$ and $\pi = \prod_{i=1}^{d} x_{[i]P}$, let $\ell$-isogeny $\phi : M_{a,b} \rightarrow M_{a',b'}$ with $\ker(\phi) = \langle P \rangle$, where $M_{a',b'}/K : b'y^2 = x^3 + a'x^2 + x$. Then,*

$$a' = (6\tilde{\sigma} - 6\sigma + a) \cdot \pi^2 \text{ and } b' = b \cdot \pi^2 \tag{1}$$

$$\phi : (x, y) \mapsto (f(x), yf'(x)), \tag{2}$$

*where $f(x) = x \prod_{i=1}^{d} (\frac{x \cdot x_{[i]P} - 1}{x - x_{[i]P}})^2$ and $f'(x)$ is its derivative.*

As mentioned earlier, because the Montgomery curve arithmetic can only be constructed with the $x$-coordinate, the function $f(x)$ is of our main interest in (2).

Let $P$ be a point on a Montgomery curve having order $\ell = 2d + 1$. Subsequently, in projective $XZ$-coordinate we express $P$ as $P = (X : Z)$, where $x = X/Z$. Let $\phi$ be an isogeny $\ell$-isogeny, where $\ker \phi = \langle P \rangle$. From the formula proposed in [12], $P' = \phi(P) = (X' : Z')$ is computed as

$$X' = X \left( \prod_{i=1}^{d} [(X - Z)(X_i + Z_i) + (X + Z)(X_i - Z_i)] \right)^2, \tag{3}$$

$$Z' = Z \left( \prod_{i=1}^{d} [(X - Z)(X_i + Z_i) - (X + Z)(X_i - Z_i)] \right)^2 \tag{4}$$

For $\ell$-isogeny evaluaton, the computational cost is $(4d)\mathbf{M} + 2\mathbf{S} + (4d + 2)\mathbf{a}$.

As denoted in (1), the computation of the image curve using Theorem 1 in [12] is somewhat complicated. Therefore, an alternate way to recover the coefficient of the image curve is presented in [12]. The first method is to use a two-torsion point of a Montgomery curve, and another is to use two

points and its differential of a Montgomery curve. We shall call the former method as a two-torsion method and the later as a differential method. As the two-torsion method is of our primary interest in this paper, we shall only describe the details of the two-torsion method in this paper. Additionally, we provide two other ways to compute the coefficient of the image curve that is presented in [7,8], in the following subsection.

**Remark 1.** *Using the differential method, we can alternatively compute the image curve coefficient with the cost* $8M + 5S + 11a$ *[12]. However, unlike SIDH, as CSIDH does not require such three points, additional point evaluation is required when this method is used. Thus, when the differential method is used, CSIDH will have inefficient speed and large key size when compared to the original method. Therefore, we exclude the use of the differential method in this paper.*

*3.2. Coefficients Computations*

3.2.1. The 2-Torsion Method

In [12], the main idea is to use two-torsion points for coefficient computation, as pushing a 2-torsion point through an odd-degree isogeny preserves their order on the image curve.

For a Montgomery curve defined over $K$, it is well-known that the two-torsion point has the following form—$(0,0), (\alpha, 0), (\alpha^{-1}, 0)$ for $\alpha \in \bar{K}$. If we know $\alpha$ of the two-torsion point on a Montgomery curve, then we can recover the coefficient of a Montgomery curve. For a given elliptic curve $M_a$, since $\alpha^3 + a\alpha^2 + \alpha = 0$, we can calculate the coefficient $a$ of $M_a$ by

$$a = -(\alpha^2 + 1)/\alpha \tag{5}$$

Let $\phi : M_a \to M_{a'}$ be an isogeny of odd-degree $\ell = 2d + 1$, and $P = (\alpha, 0)$ be a two-torsion point on $M_a$. Subsequently, it is clear that $\phi(P)$ is two-torsion point on $M_{a'}$. Using this fact, we can recover the coefficient of the image curve by first, evaluating $\phi(P)$ and obtaining the coefficient by using (5). More precisely, assuming that $\phi(P) = (\alpha', 0)$, we obtain $a' = -((\alpha')^2 + 1)/\alpha'$. In projective coordinate, let $P = (X_\alpha, Z_\alpha)$, where $\alpha = X_\alpha/Z_\alpha$. Subsequently, projective curve coefficient of the image curve is derived by using the following equation.

$$a' = (A' : C') = (X_{\alpha'}^2 + Z_{\alpha'}^2 : -X_{\alpha'}Z_{\alpha'}),$$

where $\phi(P) = (X_{\alpha'} : Z_{\alpha'})$ and $a' = A'/C'$. This computation cost is $2S + 5a$. Using the two-torsion method, the cost of calculating a coefficient of $\ell = 2d + 1$-isogeny image curve is $(4d)M + 4S + (4d + 7)a$.

**Remark 2.** *Recently, in [13], Castryck and Decru proposed the CSURF algorithm using the tweaked Montgomery curve* $M_a^t : y^2 = x^3 + ax^2 - x$ *and it is about 5.68% faster than the original CSIDH. CSURF can also use the two-torsion method because three two-torsion points are on* $M_a^t(\mathbb{F}_p)$. *If* $(\alpha, 0)$ *is a two-torsion point on a tweaked Montgomery curve* $M_a^t$ *for* $\alpha \neq 0$, *then since* $\alpha^2 + a\alpha - 1 = 0$, *we can reconstruct the tweaked Montgomery coefficient a by* $a = (A : C) = -(\alpha^2 - 1)/\alpha = (Z - X)(Z + X)/XZ$, *where* $\alpha = X/Z$. *So, we can compute an image curve coefficient by one additional point evaluation and* $2M + 2a$. *Using the two-torsion method, CSURF will might be more efficient in computing odd-degree isogeny parts.*

3.2.2. Optimization by Castryck et al.

They [7] optimize (1) to compute the coefficient of the image curve, as $\mathbb{F}_p$-rational two-torsion point does not exist for the original parameters of CSIDH.

For a point $P$ of order $\ell$ on $E$ and $k \in \{1, \cdots, \ell - 1\}$, let $(X_k : Z_k)$ be the projective $x$-coordinate of $[k]P$. Define $c_i \in \mathbb{F}_p$, such that

$$\prod_{i=1}^{\ell-1}(Z_i w + X_i) = \sum_{i=0}^{\ell-1} c_i w^i$$

as polynomials in $w$, and define $\tau, \sigma$ by

$$\tau = \prod_{i=1}^{\ell-1} \frac{X_i}{Z_i}, \ \sigma = \sum_{i=1}^{\ell-1} \left( \frac{X_i}{Z_i} - \frac{Z_i}{X_i} \right)$$

Subsequently, coefficient $(a' : 1)$ of image curve of $\ell$-isogeny with the kernel $\langle P \rangle$ is computed by

$$
\begin{aligned}
(a' : 1) &= (\tau(a - 3\sigma) : 1) \, r \\
&= (ac_0 c_{\ell-1} - 3(c_0 c_{\ell-2} - c_1 c_{\ell-1}) : c_{\ell-1}^2)
\end{aligned}
\tag{6}
$$

Using this method, the cost of calculating curve coefficient is $(6d - 2)\mathbf{M} + 3\mathbf{S} + 4\mathbf{a}$ in implementation.

### 3.2.3. Exploiting Twisted Edwards Curves

In [8], Meyer and Reith proposed a Montgomery–Edwards hybrid method for implementing CSIDH. They exploited the fact that recovering the coefficient of the image curve is more efficient on twisted Edwards curves than Montgomery curves. By using the efficiency of the birational map between Montgomery curves and twisted Edwards curves, they used Montgomery curves for scalar multiplication and isogeny evaluation and used twisted Edwards curves for recovering the coefficient of the image curve.

The outline of the process is summarized in the equation below. In the equation, $\phi$ denotes an isogeny on a twisted Edwards curve, $\iota$ denotes conversion from Montgomery to twisted Edwards curves, and $\iota^{-1}$ denotes conversion from twisted Edwards to Montgomery curves.

$$M \xrightarrow{\iota} E \xrightarrow{\psi} E' \xrightarrow{\iota^{-1}} M'$$

By composing the functions $\phi = \iota^{-1} \circ \psi \circ \iota$, one can obtain the coefficient of a Montgomery curve. Using this method, the computational cost of recovering the curve coefficient is $(2d)\mathbf{M} + 6\mathbf{S} + 6\mathbf{a} + 2c(\ell)$, where $c(\ell)$ is the cost for computing $r^\ell$ for a constant $r \in \mathbb{F}_p$. Details of this method can be found in [8].

## 4. Proposed Method

In this section, we present the optimized algorithms for CSIDH group action. First, we briefly state our motivation for this paper. The idea is to use the two-torsion method to recover the coefficient of the image curve. To use the two-torsion method in [12], we adjust the prime, so that the rational two-torsion points exist on $\mathbb{F}_p$. The CSIDH using the proposed parameter is performed on the surface. We provide two versions of our modified CSIDH, where one exchanges the two-torsion points, and the other calculates the two-torsion point for a given elliptic curve.

### 4.1. Motivation

Although there is an efficient way for computing 3- and 4- isogeny on Montgomery curves, the original formula presented in [12] for computing the coefficient of the image curve is inefficient for large odd-degree isogenies. Therefore, Costello and Hisil proposed alternate methods for computing the curve coefficient of the image curve. However, these methods unfit in the CSIDH protocol, as there

is no rational two-torsion point, nor do they use the difference of two points, as in SIDH. Hence, Castryck et al. compute the coefficient of the image curve by using (6).

On the other hand, Meyer et al. exploit the twisted Edwards curve for computing the coefficients of the image curve, as there is a simple formula for recovering the coefficient proposed by Moody and Shomow in [14]. Combining Montgomery and twisted Edwards curves, Meyer's method led to speed up of CSIDH protocol. In [15], using Edwards $w$-coordinate, Kim et al. proposed optimized isogeny formula on Edwards curves, which can be used to implement CSIDH fully on Edwards curves.

To summarize, unlike SIDH, using only Montgomery curves might be an inefficient choice for implementing CSIDH protocol. However, associated in Table 1, if the application of the two-torsion method is possible, then we can implement CSIDH entirely on Montgomery curves efficiently. Therefore, we provide the way to use the two-torsion method for computing the coefficients in CSIDH by tailoring the primes used in the base field. The proposed parameter executes CSIDH on the surface. We prove that our method also provides free and transitive group action.

**Table 1.** Computation costs of the coefficient of image curve in three methods.

| Degree | Montgomery [7] | Hybrid Method [8] | 2-Torsion Method [12] |
|:------:|:--------------:|:-----------------:|:---------------------:|
| 3 | 7 **M** | 10 **M** | 8 **M** |
| 5 | 13 **M** | 12 **M** | 12 **M** |
| 7 | 19 **M** | 14 **M** | 16 **M** |
| 11 | 31 **M** | 18 **M** | 24 **M** |
| 13 | 37 **M** | 20 **M** | 28 **M** |

Note that we assume 1**S** = 1**M** based on Table 2.

### *4.2. Proposed Method*

We define a new prime and a new base curve in order to have a rational two-torsion point other than $(0,0)$ in order to use the two-torsion method. By doing so, we can construct more efficient Montgomery-only CSIDH.

#### 4.2.1. New Parameters

Let $M_a$ be a Montgomery curve defined over finite field $\mathbb{F}_p$ where $p \equiv 3 \bmod 4$. If $E$ has a 2-torsion point on $\mathbb{F}_p$ except for $(0,0)$, then the 2-torsion subgroup $M_a(\mathbb{F}_p)[2]$ satisfy $|M_a(\mathbb{F}_p)[2]| = 4$. In this situation, the supersingular elliptic curve $M_a/\mathbb{F}_p$ is on the surface satisfying $\text{End}_{\mathbb{F}_p}(M_a) \cong \mathbb{Z}[(1+\sqrt{-p})/2]$ [13]. Note that the original CSIDH uses $p \equiv 3 \bmod 8$, so that the supersingular curve $M_a/\mathbb{F}_p$ exists on the floor satisfying $\text{End}_{\mathbb{F}_p}(M_a) \cong \mathbb{Z}[\sqrt{-p}]$. Thus, in order to have two-torsion points on $\mathbb{F}_p$, we must use a prime of the form $p \equiv 7 \bmod 8$. Following the notation presented in [13], we define the set $S_p^+ = \{a \in \mathbb{F}_p \mid y^2 = x^3 + ax^2 + x \text{ is supersingular}\}$ and the set of an elliptic curves satisfying $\text{End}_{\mathbb{F}_p}(M_a) = \mathbb{Z}[(1+\sqrt{-p})/2]$ is defined by $S_{p,\mathbb{Z}[(1+\sqrt{-p})/2]}^+ = \{a \in S_p^+ \mid \text{End}_{\mathbb{F}_p}(M_a) \cong \mathbb{Z}[(1+\sqrt{-p})/2]\}$. This set splits into two partitions, as follows.

$$S_{p,\mathbb{Z}[(1+\sqrt{-p})/2],1}^+ = \{a \in S_{p,\mathbb{Z}[(1+\sqrt{-p})/2]}^+ \mid (0,0) \notin 2M_a(\mathbb{F}_p)\},$$
$$S_{p,\mathbb{Z}[(1+\sqrt{-p})/2],2}^+ = \{a \in S_{p,\mathbb{Z}[(1+\sqrt{-p})/2]}^+ \mid (0,0) \in 2M_a(\mathbb{F}_p)\}.$$

Because $S_{p,\mathbb{Z}[(1+\sqrt{-p})/2]}^+$ consists of two orbits, the group action

$$\text{cl}(\mathcal{O}) \times S_{p,\mathbb{Z}[(1+\sqrt{-p})/2]}^+ \to S_{p,\mathbb{Z}[(1+\sqrt{-p})/2]}^+$$

is free and *not* transitive group action on $S_{p,\mathbb{Z}[(1+\sqrt{-p})/2]}^+$. In order to have transitive group action, we refer to the following lemma.

**Lemma 1.** *Let $p \equiv 7$ mod $8$ and supersingular Montgomery curve $M_a$ : $y^2 = x^3 + ax^2 + x$ be on the surface. Subsequently, there exists $P = (x, y) \in M_a(\mathbb{F}_p)$, such that $[2]P = (0,0)$ if and only if $a \pm 2$ are both square in $\mathbb{F}_p$.*

**Proof.** Because $M_a$ is on the surface, there exists a two-torsion point $(\alpha, 0) \neq (0,0)$ in $M_a(\mathbb{F}_p)$. Accordingly, $a^2 - 4$ must be square in $\mathbb{F}_p$. Subsequently, $a \pm 2$ are both square or both not square in $\mathbb{F}_p$. From $[2]P = ((X+Z)^2(X-Z)^2 : -)$ where $x = X/Z$, $[2]P = (0,0)$ if and only if $X = \pm Z$. i.e., $P = (\pm 1, -)$. Because $P$ is on the curve $M_a$, at least one of $1^3 + a \cdot 1^2 + 1 = a + 2$ and $(-1)^3 + a \cdot (-1)^2 + (-1) = a - 2$ must be square in $\mathbb{F}_p$. Therefore, $a \pm 2$ are both square in $\mathbb{F}_p$. $\square$

Using this lemma, we can prove the following theorem.

**Theorem 2.** *Let $\phi$ be an odd isogeny from $M_a$ to $M_{a'}$, where $a, a' \in S^+_{p, \mathbb{Z}[(1+\sqrt{-p})/2]}$. Subsequently*

$$a, a' \in S^+_{p, \mathbb{Z}[(1+\sqrt{-p})/2], 1} \quad or \quad a, a' \in S^+_{p, \mathbb{Z}[(1+\sqrt{-p})/2], 2}$$

**Proof.** Let $P = (X : Z)$ be a 2-torsion point in $M_a(\mathbb{F}_p)$. Afterwards, $P' = (X' : Z') = \phi(X : Z)$ is a 2-torsion point in $M_{a'}$. Since two-torsion point of the Montgomery curve is of the form $(\alpha, 0)$, $a = -(X^2 + Z^2)/XZ$, where $\alpha = X/Z$. Hence, $a \pm 2 = (X \mp Z)^2/(-XZ)$. Similarly, $a' \pm 2 = (X' \mp Z')^2/(-X'Z')$. Afterwards, squareness of $a \pm 2$ (resp. $a' \pm 2$) and $-XZ$ (resp. $-X'Z'$) is the same. Additionally, by applying (3) and (4), we can know that the squareness of $-XZ$ and $-X'Z'$ is the same. Following the proof of Lemma 1, $a \pm 2$ and $a' \pm 2$ are all squares in $\mathbb{F}_p$ or not squares in $\mathbb{F}_p$. Therefore, Theorem 2 holds by Lemma 1. $\square$

By Theorem 2, we consider free and transitive group action

$$\mathrm{cl}(\mathcal{O}) \times S^+_{p, \mathbb{Z}[(1+\sqrt{-p})/2], i} \rightarrow S^+_{p, \mathbb{Z}[(1+\sqrt{-p})/2], i} \tag{7}$$

A two-torsion point $P$ on a Montgomery curve is always of the form $(\alpha, 0)$. Since $\alpha^2 + a\alpha + 1 = 0$, $\alpha \in \mathbb{F}_p$ or $\alpha \in \mathbb{F}_{p^2}$. The initial curve of the original CSIDH is $y^2 = x^3 + x$, whose $x$-coordinate of the two-torsion point is on $\mathbb{F}_{p^2}$, extension field of $\mathbb{F}_p$. Accordingly, we need new parameters that offer 2-torsion points in $M_a(\mathbb{F}_p)$ except for $(0,0)$. The followings are those parameters.

$$p = 2^4 \cdot 3^3 \cdot 5 \cdot 7 \cdot 11^2 \cdot 13 \cdot \ldots \cdot 373 - 1 \approx 2^{510.1} \tag{8}$$

$$a = \text{0x2C36E679F542D63441367BC57EFA26639FA0EE9EA65967F55F9D9BAAE672F82}$$
$$\text{BFB429BD324D738568EF225AAA1E9F32F8056B55B9833D048EE2D99131D655918} \tag{9}$$

We use the prime $p \equiv 7 \mod 8$ and the Montgomery curve $M_a$ satisfying $|M_a(\mathbb{F}_p)[2]| = 4$. Accordingly, we can apply free and transitive group action presented in (7). Note that using the above 73 consecutive odd primes starting at 3, this parameter provides less security level than the parameters of CSIDH-512. Note that the proposed parameter in this paper is just an example parameter to apply two-torsion method on CSIDH.

**Remark 3.** *Since $((a \pm 2)/p) = -1$, this parameters correspond to $S^+_{p, \mathbb{Z}[(1+\sqrt{-p})/2], 1}$.*

4.2.2. First Method—Exchanging the Two-Torsion

The first method is to exchange two-torsion points when exchanging a curve. Alice and Bob calculate curve coefficients of image curves using a two-torsion point when computing the group action and pass it along with the image curve to each other.

Alice computes her secret isogeny $\phi_A : E \rightarrow E_A$ with her secret key $[\mathfrak{a}]$, and compute the coefficient of $E_A$ through $\phi_A(T)$. Upon receiving Bob's public key $E_B$, Alice also receives $\phi_B(T)$ in

order to compute the proceeding phase. Likewise, Bob must also receive Alice's public key $E_A$ and $\phi_A(T)$. As they need to send the image of two-torsion point in projective coordinate as well as the curve, the key size will be $3b_p$ bits, where $b_p$ is the number of bits of $p$.

### 4.2.3. Second Method—Computing the 2-Torsion

Note that when using the first method, the key size is tripled to $3b_p$ bits, where $b_p$ bits is the key size of the original CSIDH protocol. This is a huge loss as compared to a little increase in speed.

Because a two-torsion point on a Montgomery curve is of the form $(\alpha, 0)$, we can calculate $\alpha$ through solving a quadratic equation modulo $p$. Also, as $T_A = \phi_A(T)$ is a 2-torsion point in $E_A(\mathbb{F}_p)$ and $T_B = \phi_B(T)$ is a two-torsion point in $E_B(\mathbb{F}_p)$, Alice and Bob can directly calculate the two-torsion point upon the receipt of the image curve computed through each other's secret isogeny.

For $p \equiv 3 \bmod 4$, if $a$ is a quadratic residue modulo $p$, then the square root of $a$ modulo $p$ is computed by $x = a^{(p+1)/4} \bmod p$. Using this equation, we can find a two-torsion point for a given elliptic curve $E$. Also, by precomputing $2^{-1} \bmod p$, we can obtain a two-torsion point with less computation. Note that the cost of computing $a^{(p+1/4)} \bmod p$ for $a \in \mathbb{F}_p$ is very small compared to the total CSIDH algorithm. Additionally, computing the square root occurs only two times throughout the total protocol.

When the second method is used, the key size decreases to $b_p$ bits again, so we can preserve the key size and improve speed. Summing up the whole process, a class group action by computing the two-torsion point is presented in Algorithm 1. The public key validation can also be performed as in [7] for both methods.

Alternatively, one can also exchange the image of the two-torsion point in affine coordinate, instead of the coefficient of the image curve. In this case, the coefficient of the image curve can be easily recovered from the received 2-torsion point, and the key size will be decreased to $b_p$ again. However, this requires two $\mathbb{F}_p$-inversions—one for recovering the affine public key and another for computing the affine two-torsion point. Thus, there is no difference between the cost of exchanging the affine image two-torsion point and our second method, and we do not explicitly consider the case.

---

**Algorithm 1** Evaluating the class group action using the second method—Computing the two-torsion

---

**Require:** $a \in \mathbb{F}_p$ such that $M_a : y^2 = x^3 + ax^2 + x$ is supersingular curve over $\mathbb{F}_p$ and an integer
       vector $(e_1, e_2, \cdots, e_n)$ for $e_i \in [-m, m]$
**Ensure:** $a'$ such that $M_{a'} : y^2 = x^3 + a'x^2 + x$ where $M_{a'} = [\mathfrak{l}_1^{e_1} \mathfrak{l}_2^{e_2} \cdots \mathfrak{l}_n^{e_n}] M_a$
 1:   Compute a two-torsion point $T$ in $M_a(\mathbb{F}_p)$ // This step is omitted in the initial group action
 2:   **while** some $e_i \neq 0$ **do**
 3:      Sample a random point $P = (x : 1)$ where $x \in \mathbb{F}_p$
 4:      Set $s \leftarrow +1$ if there exist $y \in \mathbb{F}_p$ satisfying $y^2 = x^3 + ax^2 + x$
 5:      Otherwise, $s \leftarrow -1$
 6:      Let $S = \{i \mid e_i \neq 0, \operatorname{sign}(e_i) = s\}$
 7:      **if** $S = \varnothing$ **then**
 8:        go to line 3
 9:      **else**
10:        $k \leftarrow \prod_{i \in S} \ell_i$
11:        $Q \leftarrow [(p+1)/k]P$
12:        **for** $i \in S$ **do**
13:          $R \leftarrow [k/\ell_i]Q$
14:          **if** $R \neq \infty$ **then**
15:            Compute an isogeny $\phi : M_a \rightarrow M_{a'}$ with $\ker \phi = R$
16:            $a \leftarrow a'$, $T \leftarrow \phi(T)$, $Q \leftarrow \phi(Q)$, $k \leftarrow k/\ell_i$, $e_i \leftarrow e_i - s$
17:          **end if**
18:        **end for**
19:      **end if**
20: **end while**
21: **return** $a'$

---

## 5. Implementation

In this section, we provide the implementation results and analysis. For clear expression, we shall denote the first method as `Ours_Exchange` and the second method as `Ours_Compute`.

### 5.1. Parameter and Implementation Setup

#### 5.1.1. Parameter Setting

For implementation, we used the finite field $\mathbb{F}_p$, where $p$ is the prime presented in (8), and we used the Montgomery coefficient of the initial curve presented in (9) for both CSIDH and our methods. To make an exact comparison, we use the field operations that were implemented in [7] for both CSIDH and our methods.

For a more accurate comparison, we first measured the field operations over $\mathbb{F}_p$ to examine the ratio between each operation. To this end, each field operation was repeated $10^9$ times for $\mathbb{F}_p$. Table 2 summarizes the average cycle counts of $\mathbb{F}_p$-operations and $\frac{p+1}{4}$-power of field elements.

**Table 2.** Cycle counts of the field operations over $\mathbb{F}_p$.

|  | Addition | Subtraction | Multiplication | Squaring | $a^{(p+1)/4}$ |
|---|---|---|---|---|---|
| $p_{\text{ours}}$ | 26 | 25 | 196 | 197 | 147,965 |

#### 5.1.2. Further Modification

Let $M_a$ be a Montgomery curve. In [12], the coefficient of the Montgomery curve is presented as $(\hat{A} : \hat{C}) = (a + 2 : 4)$ instead of $(A : C) = (a : 1)$ for accelerating the doubling (DBL) and differential addition (DBL&ADD) computation. The cost of DBL&ADD decreases from $8\mathbf{M} + 4\mathbf{S} + 11\mathbf{a}$ to $8\mathbf{M} + 4\mathbf{S} + 8\mathbf{a}$ and the cost of DBL decreases from $4\mathbf{M} + 2\mathbf{S} + 7\mathbf{a}$ to $4\mathbf{M} + 2\mathbf{S} + 4\mathbf{a}$, when we used the transformed coefficient. Additionally, the cost of recovering the coefficient from a two-torsion point decreases from $2\mathbf{S} + 5\mathbf{a}$ to $2\mathbf{S} + 3\mathbf{a}$.

The original CSIDH implementation in [7] does not use this transformed coefficient. Although there is an additional cost for converting the form of the coefficients, we can save the cost of scalar multiplication in all $\ell_i$-isogeny operation. As this optimization also holds in our proposed method, we applied this technique for both CSIDH and our method. The transformations $(A : C) \leftrightarrow (\hat{A} : \hat{C})$ occurs before and after the group action, where the elliptic curve arithmetic is used.

Additionally, we noticed that the optimized point evaluation that is presented in (3) and (4) are not used in the implementation of the original CSIDH. For a reasonable comparison, we apply (3) and (4) to the original CSIDH. To summarize, by using the transformed curve coefficient and additional optimization of the point evaluation in CSIDH, the difference in the performance lies purely in the computation of recovering the curve coefficient.

### 5.2. Implementation Result

The algorithms are implemented in C language to evaluate the performance of each algorithms. All cycle counts were obtained on one core of an Intel(R) Core(TM) i7-6700 CPU @ 3.40GHz, running Ubuntu 18.04.1 LTS. For compilation, we used GNU GCC version 7.5.0 with compile option -O3 using the benchmark provided by [7]. The running time and clock cycles of the group action and the entire key exchange of the original CSIDH, `Ours_Exchange`, `Ours_Compute`, and Meyer's hybrid method are as in Table 3.

**Table 3.** Wall-clock time and clock cycles of group action and full key exchange.

| | Group Action | | Key Exchange | | |
| | Wall-Clock Time | Clock Cycles | Wall-Clock Time | Clock Cycles | Key Size |
|---|---|---|---|---|---|
| CSIDH [7] | 31.12 ms | 106,067,352 cc | 132.36 ms | 451,109,078 cc | $b_p$ bits |
| Ours_Exchange | 29.03 ms | 98,935,429 cc | 124.38 ms | 423,909,009 cc | $3b_p$ bits |
| Ours_Compute | 29.05 ms | 98,994,091 cc | 124.39 ms | 423,956,423 cc | $b_p$ bits |
| Hybrid [8] | 28.04 ms | 95,557,448 cc | 119.52 ms | 407,342,736 cc | $b_p$ bits |

Because each algorithm is implemented with a non-constant time, we report the average of one-million runs. As shown in Table 3, the group action using Ours_Compute is about 7.1% faster than the original algorithm, and the entire key exchange is about 6.4% faster than the original CSIDH. The main operation for recovering a two-torsion point is computing the $\frac{p+1}{4}$-power. The cost of recovering the two-torsion point is small compared to the cost of the entire group action, as shown in Table 2. Thus, the difference between Ours_Exchange and Ours_Compute is negligible.

Meanwhile, optimized CSIDH using twisted Edwards curves is proposed in [8,15], and using the Edwards curve is more efficient than using the two-torsion method to computing the coefficient of the image curve for higher odd-degree isogenies. However, by using the two-torsion method, we can simplify the implementation as transformations between Montgomery curves and Edwards curves are not required. Moreover, by using our method, we provide the fastest performance among the CSIDH implementation, while only using Montgomery curves.

**Remark 4.** *Recently, in [16], Bernstein et al. proposed a new odd-degree isogeny evaluation algorithm, called VeluSqrt algorithm, using only $\tilde{O}(\sqrt{\ell})$ $\mathbb{F}_p$-operations, where the $\tilde{O}$ is uniform in p. Because this algorithm impacts on evaluating isogenies only, it can be applied to all methods in Table 3.*

## 6. Conclusions

In this paper, we proposed the optimized method for improving the performance of CSIDH and provided a new parameter to use our method. We set the parameters, so that the three two-torsion points on a Montgomery curve are all in $E(\mathbb{F}_p)$. Therefore, by using a two-torsion point, we optimized the cost of computing the coefficient of the image curve of odd-degree isogeny required in the group action. When our algorithm is used, the group action is about 7.1% faster than the original CSIDH and the entire key exchange is about 6.4% faster than the original CSIDH.

As mentioned before, the proposed method in this paper is still slower than the Montgomery–Edwards hybrid method presented in [8]. However, we examined that Montgomery-only implementation is still competitive enough through various studies, like [16].

To apply our method, the prime of the base field and the initial elliptic curve must be well-selected for a target security level. If we choose the parameter, which enables applying the two-torsion method, then CSIDH will be optimized further by studying the application of two-isogeny as in [13].

**Author Contributions:** Conceptualization, D.H. and S.K.; Data curation, D.H. and K.Y.; Formal analysis, D.H.; Methodology, D.H., S.K. and Y.-H.P.; Project administration, S.K. and S.H.; Resources, D.H., K.Y. and S.H.; Software, D.H.; Supervision, S.K. and S.H.; Validation, D.H. and Y.-H.P.; Writing—original draft, D.H.; Writing—review and editing, D.H. and S.K. All authors have read and agreed to the published version of the manuscript.

**Funding:** This work was supported by the National Research Foundation of Korea(NRF) grant funded by the Korea government(MSIT) (No. NRF-2020R1A2C1011769).

**Acknowledgments:** We thank the anonymous reviewers for their useful and constructive comments.

**Conflicts of Interest:** The authors declare no conflict of interest.

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
