# Peer review of "Optimized CSIDH Implementation Using a 2-Torsion Point"

_cryptography, doi:10.3390/cryptography4030020_

Round 1
Reviewer 1 Report
The article proposes a new parameter set for CSIDH that allows for better optimizations. In particular, the authors propose a class and a specific set of parameters that allow the 2-torsion method to be used. They also provide mathematical proofs of the correctness of such an approach and show the results of an implementation of the proposed algorithm.
In general, the paper is technically correct and well organized. However, its impactfulness seems somewhat limited. Using the proposed class of parameters, it is possible to use the 2-torsion method proposed by Costello et al. to compute the coefficients of the target curve under an isogeny. In this way, it is possible to achieve a ~6% speed-up compared to the base implementation of CSIDH-512. The authors note that using a mixed Edwards/Montgomery model, it is possible to outperform this result, but the proposed approach relies on Montgomery curves only. It is not clear why this would constitute a significant benefit. Moreover, the proposed parameters allow for the optimizations used in CSURF. As reported in the paper itself, CSURF achieves a 5.68% speedup compared to CSIDH-512. Thus it is not clear why the authors are not using such optimizations (i.e. why they're optimizing CSIDH instead of CSURF) or what the effects would be in such a case. Lastly, and possibly most importantly, it is not clear whether the proposed approach offers advantages over the recently-published VeluSqrt algorithm, described in Faster Computation Of Isogenies Of Large Prime Degree by Bernstein and others. The algorithm improves image points and target curve computations and speeds up CSIDH-512 by about 8%.
The writing is generally good, clear, and mostly understandable. Some notations, however, are not introduced (such as 2M_{a}^{+} used in the equation in par. 2.4.1).
Overall, the paper is correct and provides some improvements, although their impact appears limited. Thus, I would suggest weak acceptance.
Reviewer 2 Report
This article proposes an optimisation of the CSIDH cryptosystem, resulting in an implementation that performs 6.1% faster than the original CSIDH implementation. Their optimisation consists in applying in the context of CSIDH the 2-torsion method introduced by Costello and Hisil, originally for speeding up SIDH.
The main contribution is in showing that CSIDH can be set up in a way that allows the 2-torsion method to be used. This translates into choosing a suitable prime p (resulting in a convenient F_p-rational 2-torsion), and that the resulting action of the class group on some set of Montgomery curves is free and transitive.
Understanding the performance of CSIDH and various ways to improve it is a worthwhile goal, and even though the gain in this paper is of only 6%, the methods employed deserve consideration. What is lacking from the current version is proper comparison to other optimisations of CSIDH. It is only compared (table 3) to the original "proof of concept" CSIDH implementation. Subsequent optimisations are ignored, which sounds unfair. Some are cited in the introduction [8, 9, 13]; why not discuss them in the implementation result? The compatibility between these methods and the 2-torsion method should also be discussed: are they "orthogonal" or "mutually exclusive"? I suggest to include such a discussion.
The English needs some adjustment. In the introduction:
- page 2, line 63 "various researches"
- page 3, line 90 "details of our proof are denoted"
- page 3, line 98 "a various way"
A lot of "the" are missing throughout the paper. For instance, in remark 1 (page 6), it should be "the CSURF algorithm", "the tweaked Montgomery curve", and "the tweaked Montgomery coefficient".
Some phrasings are also scientifically misleading or wrong:
- page 1, line 24 "having the same order", that is true for CSIDH but not all isogeny-base cryptography
- page 2, line 42 "SIDH is known to have exponential time complexity", better phrased as "the best known attacks against SIDH have exponential time complexity"
